# T Cell Transcriptional Signatures of Influenza A/H3N2 Antibody Response to High Dose Influenza and Adjuvanted Influenza Vaccine in Older Adults

**DOI:** 10.3390/v14122763

**Published:** 2022-12-11

**Authors:** Iana H. Haralambieva, Huy Quang Quach, Inna G. Ovsyannikova, Krista M. Goergen, Diane E. Grill, Gregory A. Poland, Richard B. Kennedy

**Affiliations:** 1Mayo Clinic Vaccine Research Group, Mayo Clinic, Rochester, MN 55905, USA; 2Department of Quantitative Health Sciences, Mayo Clinic, Rochester, MN 55905, USA

**Keywords:** influenza, influenza vaccine, hemagglutination inhibition, humoral, immunity, gene expression profiling, transcriptome, genetic markers, T cells

## Abstract

Older adults experience declining influenza vaccine-induced immunity and are at higher risk of influenza and its complications. For this reason, high dose (e.g., Fluzone) and adjuvanted (e.g., Fluad) vaccines are preferentially recommended for people age 65 years and older. However, T cell transcriptional activity shaping the humoral immune responses to Fluzone and Fluad vaccines in older adults is still poorly understood. We designed a study of 234 older adults (≥65 years old) who were randomly allocated to receive Fluzone or Fluad vaccine and provided blood samples at baseline and at Day 28 after immunization. We measured the humoral immune responses (hemagglutination inhibition/HAI antibody titer) to influenza A/H3N2 and performed mRNA-Seq transcriptional profiling in purified CD4^+^ T cells, in order to identify T cell signatures that might explain differences in humoral immune response by vaccine type. Given the large differences in formulation (higher antigen dose vs adjuvant), our hypothesis was that each vaccine elicited a distinct transcriptomic response after vaccination. Thus, the main focus of our study was to identify the differential gene expression influencing the antibody titer in the two vaccine groups. Our analyses identified three differentially expressed, functionally linked genes/proteins in CD4^+^ T cells: the calcium/calmodulin dependent serine/threonine kinase IV (CaMKIV); its regulator the TMEM38B/transmembrane protein 38B, involved in maintenance of intracellular Ca2^+^ release; and the transcriptional coactivator CBP/CREB binding protein, as regulators of transcriptional activity/function in CD4^+^ T cells that impact differences in immune response by vaccine type. Significantly enriched T cell-specific pathways/biological processes were also identified that point to the importance of genes/proteins involved in Th1/Th2 cell differentiation, IL-17 signaling, calcium signaling, Notch signaling, MAPK signaling, and regulation of TRP cation Ca2^+^ channels in humoral immunity after influenza vaccination. In summary, we identified the genes/proteins and pathways essential for cell activation and function in CD4^+^ T cells that are associated with differences in influenza vaccine-induced humoral immunity by vaccine type. These findings provide an additional mechanistic perspective for achieving protective immunity in older adults.

## 1. Introduction

Influenza causes considerable morbidity and mortality, with an annual illness burden of 9–41 million cases, leading to 140,000–710,000 hospitalizations and 12,000–52,000 deaths during the 2010–2020 time period in the United States [1]. Although the influenza virus can infect people of any age group and disease severity varies, a significantly higher risk of serious influenza complications is evident in older adults. As a result, older adults constitute 50 to 70% of influenza-associated hospitalizations and 70–85% of influenza-linked deaths [2,3,4,5]. Seasonal influenza vaccination remains the best way to prevent influenza and its complications [6]. However, older adults are often insufficiently protected by vaccination [7], due to the gradual decline of immune system response mechanisms with aging, often referred to as immunosenescence [8].

In the United States, two influenza vaccines are recommended for adults ≥65 years old: high-dose inactivated influenza vaccine (Fluzone) and adjuvanted inactivated influenza vaccine (Fluad). During the period of our study, both vaccines contained the influenza A/Michigan/45/2015 (H1N1)pdm09-like virus hemagglutinin (HA), the A/Singapore/INFIMH-16-0019/2016 A(H3N2)-like virus HA, and the B/Colorado/06/2017-like (Victoria lineage) virus HA. Each dose of the Fluzone (high-dose) vaccine contains 60 µg of HA per strain (4-fold higher than the standard dose influenza vaccine), and Fluad (adjuvanted vaccine) contains 15 µg HA (equivalent to the standard dose vaccine), formulated with MF59 adjuvant. Both the Fluzone and Fluad vaccines elicit superior antibody responses, compared to standard dose influenza vaccines [9,10,11]. The high-dose Fluzone vaccine induces higher HAI titers than the standard-dose vaccine and demonstrates better protection and a reduced risk of respiratory-related hospitalization in older adults of ≥65 years old [9,12]. Similarly, the Fluad adjuvanted vaccine was also found to be more effective in protecting older adults from pneumonia and influenza-related hospitalization, compared to the standard-dose, non-adjuvanted vaccine [13,14]. The relative effectiveness (in subjects ≥ 65 years old) of adjuvanted influenza vaccine was found to be greater, compared to the high-dose influenza vaccine, during the 2017–2018 and 2018–2019 influenza seasons [15]. Hence, a better understanding of the immunologic mechanisms driving influenza vaccine immunogenicity and efficacy is critical not only to enhance vaccine effectiveness, but also to design better influenza vaccines for older individuals. Herein, we comprehensively investigate the transcriptional and immune responses to the Fluzone and Fluad vaccines in 234 adults ≥ 65 years old and identify differential transcriptional immune signatures in CD4^+^ T cells, underlying the differences in antibody titers in vaccine recipients of the Fluzone and Fluad vaccines. This work sheds further light on mechanistic explanations, regarding the the observed differences in vaccine-induced humoral immunity (influenza A/H3N2 HAI antibody titer) by vaccine type.

## 2. Methods

The following methods are similar or identical to our previously published studies [16,17,18,19,20].

### 2.1. Study Subjects

This study was approved by The Mayo Clinic Institutional Review Board (IRB# 17-010601).

During the period of August to December 2018, we enrolled 234 healthy subjects, aged ≥ 65 years, who were randomly allocated to receive either high-dose influenza vaccine (Fluzone) or MF59-adjuvanted influenza vaccine (Fluad). Both inactivated vaccines contained antigens of the: (i) H1N1 (A/Michigan/45/2015/pdm09-like strain), (ii) H3N2 (A/Singapore/INFIMH-16-0019/2016-like strain), and (iii) B lineage (Colorado/06/2017-like Victoria lineage strain) strains. Blood draws were obtained on all subjects at baseline (prior to vaccination) and 28 days following vaccination. Written informed consent was obtained from each subject at the time of enrollment. All subjects were asked to provide their vaccination histories and report their health status. During the study, study subjects were excluded if they exhibited symptoms related to influenza.

### 2.2. Processing Blood Samples

Blood samples were collected and processed as previously published [19]. Briefly, blood (100 mL) was collected from each subject at two different time points: day 0 (baseline before immunization) and day 28 after immunization (peak of humoral immune responses). Each blood sample included: (i) 90 mL of blood collected in 9 BD Vacutainer^®^ tubes containing lithium heparin (anticoagulant) that was used for isolation of peripheral bone mononuclear cells (PBMCs), and (ii) 10 mL of blood collected in 2 BD Vacutainer^®^ tubes (containing clot activator) that was used for the preparation of serum. PBMCs were isolated by gradient centrifugation using Ficoll Paque Plus tube (GE Healthcare Life Sciences, Uppsala, Sweden), following the manufacturer’s instructions. Isolated PBMCs were resuspended in freezing media (20% heat-inactivated FCS, 10% DMSO in RPMI) at 1 × 10^7^ cells/mL and stored in liquid nitrogen for future use. For serum, blood tubes were centrifuged at 2000× *g* for 15 min at room temperature. Serum was collected from the supernatant after centrifugation and stored at −80 °C.

### 2.3. Influenza Hemagglutination Inhibition (HAI) Assay

Hemagglutination inhibition (HAI) assays were performed using the influenza A/H3N2 virus (A/Singapore/INFIMH-16-0019/2016/H3N2 strain), in accordance with WHO guidelines [20] and following an optimized protocol, as previously published [17]. Briefly, sera were treated with receptor-destroying enzyme (RDE) from *Vibrio cholerae* filtrate (Sigma-Aldrich, St. Louis, MO, USA) to eliminate non-specific inhibitors of hemagglutination. Treated sera were two-fold serially diluted with PBS in a V-bottom 96-well plate. Diluted sera (25 µL) were mixed with 25 µL of standardized influenza A/H3N2 virus solution containing 4 hemagglutination units (HAU) and incubated for 30 min at room temperature. Then, 0.5% turkey red blood cell suspension (50 µL) was added to the mixture. The HAI titer were determined after 45 min incubation and defined as the reciprocal of the highest serum dilution that completely inhibited hemagglutination. All dilutions were tested in triplicate. A positive pooled control serum was run with each batch of samples. The coefficient of variation (CV) of the HAI assay in our laboratory was 2.9% [21].

### 2.4. mRNA Sequencing

mRNA next-generation sequencing was performed using similar protocols, as described in our previous publications [16,18,22]. Briefly, untouched CD4^+^ Tcells were first isolated from PBMCs via negative selection (depletion of non-target cells expressing CD8, CD14, CD15, CD16, CD19, CD34, CD36, CD56, CD123, and CD235a), using the Miltenyi Biotec’s Pan-T cell isolation kit, CD8 specific magnetic microbeads, LS columns, and a MidiMACS™ Separator. Total RNA was extracted from isolated cells using the RNeasy Plus Mini Kit and RNAprotect reagent (Qiagen, Valencia, CA, USA). RNA quantity and quality were assessed by Nanodrop (Thermo Fisher Scientific, Wilmington, DE, USA) and Agilent 2010 Bioanalyzer (Agilent, Palo Alto, CA, USA) assays. The TruSeq^®^ Stranded mRNA Library Prep v2 (Illumina, San Diego, CA, USA) was used to create cDNA libraries, following the manufacturer’s protocol at Mayo Clinic’s Gene Sequencing Core Facility. Illumina’s NovaSeq 6000 S2 Reagent Kit (100 cycles) was used to perform paired-end read sequencing on the Illumina NovaSeq 6000 Instrument. The MAP-RSeq version 3.0 pipeline was used to align reads using STAR to the hg38 human reference genome, and gene expression counts were obtained using featureCounts, utilizing the gene definition files from Ensembl v78 [23]. Conditional quantile regression was used for normalization [24]. All gene expression values within each gene were scaled to have a mean of zero and a standard deviation of one prior to analyses.

### 2.5. Statistical Analysis

Differences between vaccine types were tested with Wilcoxon rank sum test for continuous variables and Pearson’s chi-squared test for discrete variables. To assess the effect of both gene expression and vaccine type on the immune outcome, we fit linear regression models with each gene and immune response variable. The dependent variables in the models were scaled gene expression, vaccine type, and the interaction between vaccine type and gene expression (i.e., interaction model). An interaction model with a significant interaction term/finding indicates that the predicted immune response trajectory is different for each vaccine for the specified gene. Interaction plots are presented, showing the predicted immune response separately for each vaccine type across the scaled gene expression. In all our models, the immune outcome was defined as the log2 influenza A/H3N2 HAI titer (Day 28–Day 0). Due to the large number of tests, false discovery rate q-values (for the per gene comparison analyses) were calculated using Storey’s method [25]. In the per-gene analyses, we set significance thresholds at *p* ≤ 0.005/q ≤ 0.2. Gene set enrichment analysis was conducted using the ClusterProfiler package in R, and the online KEGG module and pathway data sets [26]. We restricted the geneset size to be between 3 and 800. FDR for the pathway analysis were calculated using Benjamini–Hochberg method [27].

## 3. Results

### 3.1. Demographic and Immune Response Characteristics for the Study Cohort

Of 250 subjects initially enrolled for this study, 9 subjects were excluded due to the lack of vaccination history and 7 subjects withdrew during the study. Of the 234 subjects who completed all study visits for blood sampling, 146 (62.4%) were females and 88 (37.6%) were males (Figure 1, Table 1). The racial/ethnicity characteristics of the study subjects reflected the Olmsted County population, with Hispanics and African Americans constituting only 1.7% and 0.4% of the study subjects, respectively (Table 1). An equal number of subjects (n = 117) were randomly allocated to receive either high-dose Fluzone or MF59-adjuvated Fluad vaccine (Table 1). Demographic variables were comparable between the two groups, including height, weight, and BMI measurements (Table 1). In our previous studies, demographic variables have been found to significantly impact vaccine-induced immune responses [8,19,28]. Since demographic/clinical variables in this study were not significantly different between our two vaccine groups, this allowed us to investigate the impact of the influenza vaccine type on immune responses, without these confounding differences between groups. Both vaccines were immunogenic and resulted in a rise in influenza A/H3N2 HAI antibody titer, relative to the baseline (Appendix A), with no statistically significant differences in antibody response between males and females (data not shown). The HAI Ab titer change against influenza A/H3N2 virus (log_2_ change in HAI titer from Day 0 to Day 28, Table 1, Appendix A) demonstrated a slightly higher response in the Fluzone vaccine recipients, although this difference did not reach statistical significance between the two cohorts (*p* = 0.06, Table 1).

### 3.2. Humoral Immune Response to Fluad and Fluzone Vaccination Is Shaped by Differential CD4^+^ T Cell Gene Expression

We tested the hypothesis that vaccine-induced immunity/immunogenicity occurs through different activation of T cell help by vaccine type, and these differences are influenced by gene expression in the CD4^+^ T cell population. To do this, we conducted three interaction modeling analyses, allowing for the effect of gene expression to vary by vaccine type, in order to assess the effect on immune outcome. These included: interaction model 1, where we assessed the effect of the Day 28/Day 0 gene expression change; interaction model 2, where we assessed the effect of the baseline/Day 0 gene expression; and interaction model 3, where we assessed the effect of Day 28 gene expression. The immune outcome in all analyses is defined as the log2 influenza A/H3N2 HAI titer (Day 28–Day 0). Significance thresholds at *p* ≤ 0.005/q ≤ 0.2 were used in these analyses.

First, we explored the influence of CD4^+^ T cell transcriptomic changes over time (Day 28/Day 0 gene expression changes—interaction model 1) on HAI antibody titer following vaccination. This model identified 128 genes that meet our significance criteria (Table 2, Appendix A). The majority of these genes/factors have no known direct immune function, but are involved in cellular processes in T cells, such as cell signaling, metabolism, transcription, and membrane trafficking, with a plausible or established link to immune processes (Table 2, Appendix A). Among our major significant findings, we identified *CAMK4*, which encodes the calcium-/calmodulin-dependent serine/threonine kinase IV (CaMKIV), (*p* = 0.0008, q = 0.1575, Table 2). Another significant gene of interest (*TMEM38B*) is linked to CAMKIV function and encodes a cation channel protein involved in the regulation of intracellular calcium [29]. These observations are consistent with the reported role of CAMKIV in the development of humoral immunity after influenza vaccination in previous system vaccinology studies [30,31,32]. Similar to these previous reports, we found that *CAMK4* gene expression was negatively associated with antibody titer in our overall study cohort (gene estimate −0.4262, *p* = 0.0006, data not shown), but also demonstrated that this gene/factor is likely involved in the differential activation of T cell help by vaccine type (Figure 2). Figure 2 visualizes the differential effect of *CAMK4* and several functionally-related genes on the immune outcome in the two vaccine groups. These significant genes were selected for display from interaction model 1, because they were functionally linked and previously reported (for *CAMK4*) to influence immune responses following influenza vaccination [30,31,32]. The negative association of the *CAMK4* gene expression change with the immune outcome was quite pronounced for the Fluad vaccine. Conversely, there was a very slight positive association for the Fluzone vaccine (Figure 2). Of note, similar differential patterns were observed for other important genes, as well (Figure 2). Slightly below our significance/false discovery rate thresholds (with *p* = 0.0086, q = 0.2562, Table 2), we also identified a third CAMKIV-linked gene/factor, the *CREBBP*/CREB binding protein (CBP), which is a transcriptional coactivator of CREB (a master regulator of transcription in T cells) regulated by nuclear Ca^2+^ and CaMKIV [33].

Among the top 20 genes (Table 2), we also identified four members of the solute carrier family proteins (*SLC2A11*, *SLC38A5*, *SLC25A53*, *SLC23A3*) that are metabolic regulators of T cell differentiation and function [29] and have a role in innate and inflammatory responses in macrophages [34,35]. Other interesting genes with a role in T cell differentiation/activation/function from interaction model 1 include: *PLEKHG5*/pleckstrin homology and RhoGEF domain containing G5 (a protein that activates the NF-kappa-B signaling pathway); *PIAS3*/protein inhibitor of activated STAT 3; *TRAF7*/TNF receptor associated factor 7; *GATA3*/GATA binding protein 3 (transcriptional regulator of Th2 differentiation/function); and IL36A/interleukin 36 alpha (a protein that can activate NF-kappa-B and MAPK signaling pathways, involved in inflammation (Table 2). All significant genes from interaction model 1 are listed in Appendix A.

Second, we explored the interaction of the genes by vaccine type, for Day 0 (baseline) gene expression (interaction model 2) or Day 28 gene expression (interaction model 3) on the antibody response/HAI titer (Day 28–Day 0) (Table 3). Although we observed a few interesting immune signaling and regulatory factors among our top findings from these models (i.e., *PIAS3*/protein inhibitor of activated STAT 3 (also identified in interaction model 1, described above); *NLRX1*/NLR family member X1, a versatile immune regulatory protein [36]; and *POGLUT1*/ protein O-glucosyltransferase 1, an important regulator of Notch signaling, see Table 3), the resultant findings did not meet our statistical threshold and must be interpreted with caution.

### 3.3. Biological Pathways and Immune Functions in T Cells Associated with the Immune Response by Vaccine Type

To better understand the biological processes and pathways controlled by the differentially expressed genes in CD4^+^ T cells between the Fluad and Fluzone vaccine recipients, we performed pathway enrichment analysis on all the genes from interaction model 1. This analysis identified pathways/biological functions that are significantly represented in our results, capturing the effects of genes/factors below the significance threshold. We used pathway definitions from MSigDB’s index of KEGG canonical pathways (see Statistical Methods) [37]. In the enrichment analysis, we used a significance/false discovery rate threshold of q ≤ 0.05 (Table 4) for pathway identification. We found 42 significantly enriched pathways, of which 7 have been associated with non-influenza viral infections and mostly included overlapping inflammatory and innate genes (Table 4). A total of 11 of the 42 significantly enriched pathways were directly related to T cell differentiation, activation, and/or function, with a variety of signal transduction, transcription-related, effector, and other genes involved in Th1/Th2 cell differentiation, TNF signaling pathway, Notch signaling pathway, MAPK signaling pathway, IL-17 signaling pathway, calcium signaling pathway, regulation of transient receptor potential (TRP) cation channels (stress-sensitive Ca^2+^-permeable ion channels), rap1/GTPase signaling pathway (important for the interaction between T cells and APCs, finetuning TCR signaling, and T cell responses) [38], and tight junction and apoptosis (Table 4, T cell-related pathways highlighted in bold). Significantly enriched pathways based on genes from interaction models 2 and 3 are presented in Appendix A. Pathways linked to T cell activity, such as Th1/Th2 cell differentiation, Th17 cell differentiation, and multiple signaling pathways, were also found to be enriched (Appendix A).

## 4. Discussion

High-dose Fluzone and adjuvanted Fluad vaccines are known to induce a more robust humoral immune response in older adults than conventional influenza vaccines, with a consequent increase in protection against influenza. However, little is known about T cell transcriptional signatures shaping the humoral response to these vaccines in older adults, who are at risk of immunosenescence-related decrease in immune function (i.e., have suboptimal vaccine-induced immunity) and at higher risk of influenza and influenza-linked complications. The major focus of our study was to identify transcriptional immune signatures in the T helper cell compartment and the possible mechanisms underlying differences in vaccine-induced humoral immunity between the Fluzone and Fluad vaccines in older adults (≥65 years old). We hypothesized that T cell signaling and function play an essential role in those differences and employed an immune response modeling approach using the interaction of CD4^+^ T cell gene expression by vaccine type.

The goal of our research efforts was to find T cell genes differentially influencing the immune response/antibody titer in the two vaccine groups. Our per-gene modeling efforts identified at least three functionally-linked proteins, i.e., the multifunctional CaMKIV kinase, one of its regulators/activators (transmembrane protein 38B, involved in maintenance of intracellular Ca2^+^ release), and its immediate interacting partner (the CREB binding protein/CBP), as pivotal players in the T-cell signaling events and transcriptional regulation in CD4^+^ T cells, thus resulting in antigen-specific activation of T cells and differential functional activity impacting humoral immunity. Importantly, we did observe a differential effect of other identified genes in each vaccine group, which significantly influenced the HAI antibody titer (Figure 2), as expected from our appied modeling strategies.

A different approach (microarray-based gene expression on RNA extracted from PBMCs) has been previously used to compare the immune responses in subjects immunized with either trivalent inactivated influenza vaccine or live attenuated influenza vaccine and led to the discovery of the role of CaMKIV kinase (gene expression at day 3) in the negative regulation of influenza vaccine humoral immunity [30,31,32]. The role of CaMKIV in influenza vaccine immunity was validated in vivo, in a CaMKIV-deficient mouse model, in which influenza vaccination led to higher antibody titers [30]. Furthermore, *CAMK4* was identified as one of the top genes of a geneset/cluster (linked to T cell activation) that was associated with reduced cellular immune responses (IFN-γ, IL-2 cytokine secretion) after influenza vaccination in 138 healthy older adults [39]. Gene signatures (derived from gene expression in PBMCs) linked to innate immune function (such as the Toll-like receptor-linked immune signature, inflammatory monocyte-linked signature, antiviral interferon signature, and dendritic cell function-linked signature) were found to be correlated with HAI antibody titer at day 28, when comparing trivalent inactivated influenza vaccine with MF59-adjuvanted trivalent inactivated vaccine in 14- to 24-months old children [40]. Importantly, the MF59-adjuvanted vaccine was demonstrated to induce a higher frequency of antigen-specific polyfunctional CD4^+^ T cells in the latter study [40], although the underlying mechanism and/or link to humoral immunity remains unknown. Our study identified pathways and functions linked to adaptive immunity/T cell immune fuinction, rather than innate immunity, (e.g., T cell differentiation, T cell activation, and T cell-linked signaling) to influnece the antibody response induced by MF59-adjuvanted vs. high-dose influenza vaccine in older adults. This suggests that differences of induced immune responses to these vaccines may vary by age, most likely due to differences in immune system components/function and immune memory in different age groups (older adults vs. very young children). Furthermore, non-specific vaccine components, such as adjuvants, may have differential effect on immune function dependent on age.

Our findings (CAMKIV, TMEM38B, CBP) are functionally linked to the downstream regulation/activation of the *CREB* gene, which encodes for the cAMP-responsive element binding protein (CREB), a transcription factor involved in different cellular responses, such as cell proliferation, cell survival, cell differentiation, and effector functions [41]. It is often triggered by extracellular stimuli (e.g., glutamate, growth, and inflammatory stimuli), activated by phosphorylation to initiate, carry forward, and facilitate transcriptional programmes of immune genes (among other genes) containing cAMP-responsive elements. Many T cell function-specific genes also contain cAMP-responsive elements (e.g., *IL2*, *IL10*, *TNF*, *TCRA/TCRα*, *TCRB/TCR Vβ*, *CD3D/CD3δ*, *CD8A/CD8α*, *CD25/IL2RA*, *IL2RG*), suggesting that CREB is a master regulator of transcriptional activity in T cells, with a major role in Th1, Th2, and Th17 differentiation, the generation/maintenance of Tregs, and T cell effector function [41]. Its role in T cells is illuminated in transgenic CREB mutant mice with dominant negative/transcriptionally inactive CREB, where defects in T cell proliferation, T cell function (IL-2 production), and apoptosis were demonstrated [42]. Similar models demonstrate defects in T helper effector function (IL-4 and IFN-γ production), with failure to mount an antigen-specific antibody response [43]. Recent findings in murine models also point to the instrumental role of CAMKIV for the expansion of Tfh lymphocytes and for the regulation of humoral immune responses through control over the *BCL6* gene expression [44].

The optimal activation/phosphorylation and transcription factor binding of CREB during T cell activation requires (in addition to the activation of the kinase cascade PKA, PKC, MAPK ERK1/2, p38, and MSK1/2 [41,45,46]) the engagement of both CD3 and CD28, as well as the recruitment/binding of the CREB-binding protein/CBP to CREB, which, in turn, is thoroughly dependent on p38 MAPK and CaMKIV activation [46]. In addition, both kinases are engaged in CREB phosphorylation, and CaMKIV alone is able to activate/phosphorylate CBP (controlled in addition by nuclear Ca2^+^) at Ser301 [33,46,47]. Thus, CAMKIV and its functionally linked partners (CBP and transmembrane protein 38B) are instrumental for the CREB-based transcriptional activity in T cells.

CaMKIV kinase signals via the calcium-dependent CaMKK-CaMK4 cascade and is activated upon increase of the intracellular Ca^2+^ concentration (controlled by proteins such as the transmembrane protein 38B/TMEM38B) and calmodulin. As described above, its function/expression is indispensable for the optimal stimulation of CREB/CBP-activated transcriptional programs in T cells, and in particular, CD4^+^ T cells with a memory phenotype [48]. In addition to the above-mentioned downstream effects, transcription factor CREB is essential for innate immune function through the inhibition of NF-kB activation in monocytes/macrophages to limit inflammation and promote survival, as well as for survival of dendritic cells upon TLR stimulation [41,49].

Our results also emphasize the significance of other T cell-specific transcription factors from our per-gene interaction model 1, such as the GATA binding protein 3, a major regulator of Th2 T cell differentiation, which regulates the expression of Th2 cytokines [50], and/or other regulatory factors and signal transducers involved in MAPK and NF-kB signaling, Notch signaling, Jak-STAT signaling, and TNF signaling. Consistent with this, our pathway enrichment analysis points to most of these pathways and biological processes and, in particular, to pathway/biological functions linked to the CaMKIV-CBP-CREB activation and downstream effects in T cells (calcium signaling pathway, regulation of TRP Ca2^+^-permeable cation channels, IL-17 signaling pathway, and Th1/Th2 differentiation, among others).

The strengths of our study include the randomized study design, as well as the use of a high-dimensional dataset linked to a specific cell type—purified CD4^+^ T cells. Of the three models we used in our analysis approach, the most significant and biologically relevant findings come from interaction model 1, where we assessed the effect of transcriptomic changes over time (rather than baseline/Day 0 and Day 28 gene expression).

The major limitation is the possibility of false positive findings, which is addressed by using *p*-value/q-value thresholds. The potential for type II error also exists, given the relatively limited numbers of individuals studied, and our results may or may not be generalizable to other races and/or ethnicities, other than the population we studied. Regardless, our analytical approach allowed us to identify transcriptomic signatures that modify vaccine-induced immunity of the two influenza vaccines we studied, leading to a better biologic understanding. The differential effect of some of the identified genes (Figure 2) on immune function/outcome in the two vaccine groups is unclear mechanistically, but could be linked to the differential trigerring (vs. no influence for example) of specific immune pathways (e.g., calcium signaling and Ca^2+^ signaling-linked T cell activation) by an adjuvant or higher antigen dose. For example, it is demonstrated that MF59 induces the release of extracellular ATP, which, in turn, can influence the transport of Ca^2+^ against elecrochemical gradient (which is ATP-dependent) and, thus, have an impact on calcium signaling and T cell activation [51,52]. An important limitation in our study is the lack of functional validation of our major targets and pathways, which is planned for future studies.

Collectively, our study identified important T cell factors and functions regulating influenza vaccine humoral immunity by vaccine type and highlinging possible differences in their mechanism of action. Further validation and new analyses regarding these unique data are in progress, including the modeling of cellular immune response outcomes following vaccination.

## Figures and Tables

**Figure 1 viruses-14-02763-f001:**
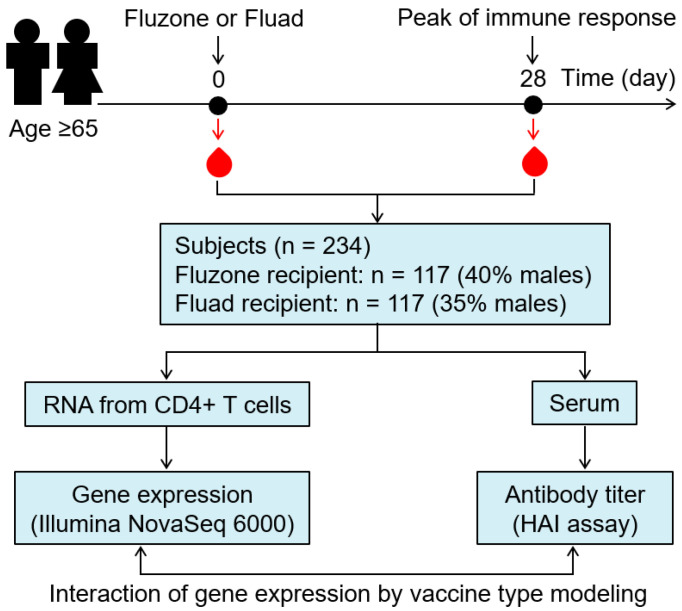
**Study design.** Of the 234 subjects enrolled (aged ≥65), an equal number (n = 117) of subjects was randomly allocated to receive either high-dose Fluzone or MF59-adjuvanted Fluad influenza vaccine. Blood was sampled before immunization (Day 0/baseline) and 28 days after immunization (Day 28). Humoral immune responses against the influenza A/H3N2 virus were assessed in serum using the hemagglutination inhibition assay (HAI). Gene expression was profiled in purified CD4^+^ T cells isolated from PBMCs. The effect of the interaction of CD4^+^ T cell gene expression by vaccine type on antibody titer was evaluated using linear regression models (see Methods).

**Figure 2 viruses-14-02763-f002:**
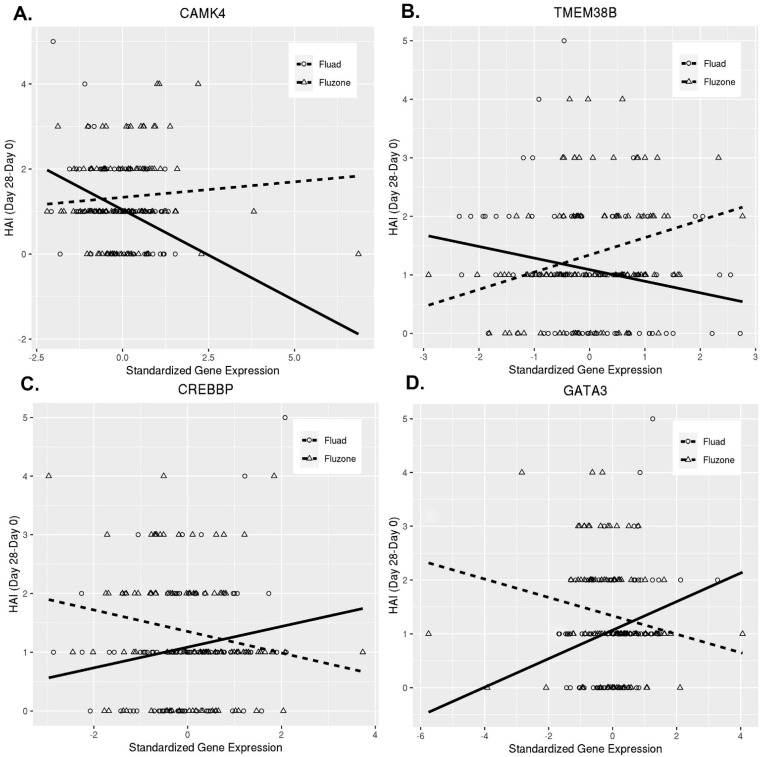
**Differential effect of selected significant genes on immune outcome in the two vaccine groups.** Interaction plots for *CAMK4* (**A**), *TMEM38B* (**B**), *CREBBP* (**C**), and *GATA3* gene (**D**) demonstrate the differential effect on the immune outcome by vaccine type (i.e., the different trajectories of association for the Fluad (solid line, circles) versus Fluzone (dashed line, triangles) vaccines for the specific genes. The horizontal axis represents the standardized gene expression Day 28/Day 0 change (gene expression minus the mean divided by the standard deviation); a one unit change in the standard gene expression is equivalent to a one standard deviation change in gene expression. The vertical axis is the log_2_ change in HAI titer from Day 0 to Day 28; a one unit increase in the log2 HAI measure represents a doubling of the HAI titer from Day 0 to Day 28.

**Table 1 viruses-14-02763-t001:** Demographic and clinical variables of the study cohort.

Variables	Fluad (n = 117)	Fluzone (n = 117)	Total (n = 234)	*p*-Value
Sex				0.418 *
Female	76 (65.0%)	70 (59.8%)	146 (62.4%)	
Male	41 (35.0%)	47 (40.2%)	88 (37.6%)	
Age at enrollment (years)				0.556 **
Median	71.900	71.100	71.500	
Q1, Q3	68.200, 76.500	67.700, 75.600	67.900, 76.000	
Ethnicity				1.000 *
Non-Hispanic or Latino	115 (98.3%)	115 (98.3%)	230 (98.3%)	
Hispanic or Latino	2 (1.7%)	2 (1.7%)	4 (1.7%)	
Race				0.316 *
Black or African American	1 (0.9%)	0 (0.0%)	1 (0.4%)	
White	116 (99.1%)	117 (100.0%)	233 (99.6%)	
Weight (kg)				0.977 **
Median	79.5	81.1	80.3	
Q1, Q3	68.9, 91.1	68.1, 90.7	68.5, 91.0	
Height (cm)				0.221 **
Median	164.4	165.9	164.9	
Q1, Q3	159.1, 173.5	161.2, 175.2	160.1, 174.4	
BMI				0.333 **
Median	28.7	28.0	28.3	
Q1, Q3	25.3, 31.9	24.9, 31.1	25.2, 31.6	
HAI titer (Day 28/Day 0) ***				0.062 **
Median	1.000	1.000	1.000	
Q1, Q3	0.000, 2.000	1.000, 2.000	0.000, 2.000	

* Pearson’s Chi-squared test. ** Wilcoxon rank sum test. *** Influenza A/H3N2 HAI antibody titer (log_2_ change in HAI titer from Day 0 to Day 28).

**Table 2 viruses-14-02763-t002:** Influence of vaccine type and CD4^+^ T cell gene expression change on Ab response to influenza vaccination (Day 28-Day 0 HAI titer change). Full list of genes is expanded in Appendix A.

Day 28/Day 0 Gene Expression Change (Interaction Model 1) *				
Gene Symbol	Entrezgene Description	Estimate	*p*-Value **	q-Value ***
*SLC2A11*	solute carrier family 2 member 11	0.6034	8.19 × 10^−6^	0.0701
*RP11-779O18.2*	pre-mRNA processing factor 31	−0.5604	2.65 × 10^−5^	0.1137
*SLC38A5*	solute carrier family 38 member 5	−0.5324	0.0001	0.1575
*WDR35*	WD repeat domain 35	0.5332	0.0001	0.1575
*APOBR*	apolipoprotein B receptor	−0.5347	0.0001	0.1575
*CC2D1A*	coiled-coil and C2 domain containing 1A	−0.4902	0.0002	0.1575
*MYO9B*	myosin IXB	−0.5059	0.0002	0.1575
*PRKD2*	protein kinase D2	−0.5016	0.0002	0.1575
*SLC25A53*	solute carrier family 25 member 53	0.4843	0.0003	0.1575
*TMEM38B*	transmembrane protein 38B	0.4931	0.0003	0.1575
*B4GALT4*	beta-1,4-galactosyltransferase 4	0.4931	0.0003	0.1575
*SLC23A3*	solute carrier family 23 member 3	0.4830	0.0003	0.1575
*GORASP1*	golgi reassembly stacking protein 1	0.4830	0.0004	0.1575
*ERLIN2*	ER lipid raft associated 2	0.4828	0.0004	0.1575
*PITPNM1*	phosphatidylinositol transfer protein membrane-associated 1	−0.4753	0.0004	0.1575
*MAP4*	microtubule associated protein 4	−0.4993	0.0004	0.1575
*MVB12B*	multivesicular body subunit 12B	−0.4904	0.0004	0.1575
*PKIG*	cAMP-dependent protein kinase inhibitor gamma	0.4634	0.0005	0.1575
*POP5*	POP5 homolog, ribonuclease P/MRP subunit	0.4947	0.0005	0.1575
*C17orf51*	long intergenic non-protein coding RNA 2693	0.4690	0.0005	0.1575
*PLEKHG5*	pleckstrin homology and RhoGEF domain containing G5	−0.4809	0.0005	0.1575
*CAMK4*	calcium-/calmodulin-dependent protein kinase IV	0.4991	0.0008	0.1575
*PIAS3*	protein inhibitor of activated STAT 3	−0.4458	0.0009	0.1598
*TRAF7*	TNF receptor-associated factor 7	−0.4395	0.0012	0.1684
*COX6B1*	cytochrome c oxidase subunit 6B1	0.4254	0.0017	0.1783
*GATA3*	GATA-binding protein 3	−0.4358	0.0019	0.1861
*IL36A*	interleukin 36 alpha	0.4159	0.0024	0.1967
*MAP3K13*	mitogen-activated protein kinase 13	0.4208	0.0025	0.1995
*CREBBP*	CREB-binding protein	−0.3594	0.0086	0.2562

* Top 20 genes from the model are presented, followed by other significant genes/finding (below the line), with a link to T cell differentiation/activation/function, resulting from this model. ** *p*-values are from the interaction term of the linear model. *** Q-values were calculated using Storey’s method (see Methods).

**Table 3 viruses-14-02763-t003:** Influence of CD4+ T cell gene expression (Day 0 or Day 28) on Ab response to influenza vaccination by vaccine type.

**Baseline (Day 0) Gene Expression (Interaction Model 2) ***				
**Gene Symbol**	**Entrezgene Description**	**Estimate**	***p*-Value ****	**q-Value *****
*FAM136A*	family with sequence similarity 136 member A	0.5098	0.0002	0.4912
*POLA1*	DNA polymerase alpha 1, catalytic subunit	−0.4990	0.0002	0.4912
*CCER2*	coiled-coil glutamate rich protein 2	0.4806	0.0004	0.4912
*MLLT4/AFDN*	afadin, adherens junction formation factor	0.4812	0.0004	0.4912
*AMY2B*	amylase alpha 2B	0.4763	0.0004	0.4912
*HECTD3*	HECT domain E3 ubiquitin protein ligase 3	0.4935	0.0005	0.4912
*C2orf66*	chromosome 2 open reading frame 66	0.4616	0.0006	0.4912
*PIGW*	phosphatidylinositol glycan anchor biosynthesis class W	0.4605	0.0006	0.4912
*SLC23A3*	solute carrier family 23 member 3	−0.4616	0.0006	0.4912
*BAIAP3*	BAI1-associated protein 3	0.4622	0.0007	0.4912
*ITIH4*	inter-alpha-trypsin inhibitor heavy chain 4	0.4559	0.0007	0.4912
*MUSTN1*	musculoskeletal, embryonic nuclear protein 1	0.4587	0.0007	0.4912
*MBTPS2*	membrane bound transcription factor peptidase, site 2	−0.4660	0.0008	0.4912
*MYO19*	myosin XIX	0.4509	0.0008	0.4912
*NLRX1*	NLR family member X1	−0.4588	0.0009	0.4912
*RP5-966M1.6*	SREBF2 antisense RNA 1	0.4501	0.0009	0.4912
*PIAS3*	protein inhibitor of activated STAT 3	0.4545	0.0009	0.4912
*C14orf119*	chromosome 14 open reading frame 119	−0.4441	0.0010	0.4912
*ALDH8A1*	aldehyde dehydrogenase 8 family member A1	0.4435	0.0011	0.4912
*MRPS28*	mitochondrial ribosomal protein S28	−0.4512	0.0012	0.4912
**Day 28 Gene Expression (Interaction Model 3) ***				
**Gene Symbol**	**Entrezgene Description**	**Estimate**	***p*-Value ****	**q-Value *****
*COQ4*	coenzyme Q4	0.5698	2.15 × 10^−5^	0.2293
*EEFSEC*	eukaryotic elongation factor, selenocysteine tRNA-specific	−0.5304	0.0001	0.3727
*NEIL1*	nei-like DNA glycosylase 1	0.5243	0.0001	0.3727
*SLC25A14*	solute carrier family 25 member 14	0.5000	0.0002	0.4405
*CNTD2/CCNP*	cyclin P	0.4896	0.0003	0.4405
*PSMC3IP*	PSMC3 interacting protein	0.4946	0.0003	0.4405
*PPM1N*	protein phosphatase, Mg2+/Mn2+-dependent 1N	0.4857	0.0003	0.4405
*MYO19*	myosin XIX	0.4800	0.0003	0.4405
*BAIAP3*	BAI1-associated protein 3	0.4668	0.0007	0.7302
*BIVM*	basic, immunoglobulin-like variable motif containing	0.4614	0.0008	0.7302
*ECHDC2*	enoyl-CoA hydratase domain containing 2	0.4586	0.0008	0.7302
*TTC8*	tetratricopeptide repeat domain 8	0.4493	0.0008	0.7364
*POGLUT1*	protein O-glucosyltransferase 1	0.4451	0.0009	0.7598
*TRIP6*	thyroid hormone receptor interactor 6	0.4316	0.0011	0.7658
*SAPCD1*	suppressor APC domain containing 1	0.4472	0.0012	0.7658
*CYP4F12*	cytochrome P450 family 4 subfamily F member 12	0.4541	0.0014	0.7658
*MFI2/MELTF*	melanotransferrin	0.4405	0.0014	0.7658
*LPPR2/PLPPR2*	phospholipid phosphatase-related 2	0.4282	0.0014	0.7658
*FN3KRP*	fructosamine 3 kinase-related protein	0.4198	0.0016	0.7658
*RABL2A*	RAB, member of RAS oncogene family-like 2A	0.4191	0.0016	0.7658

* Top 20 genes from the model are presented. ** *p*-values are from the interaction term of the linear model. *** Q-values were calculated using the Storey’s method (see Methods). In summary, our most significant and biologically meaningful transcriptomic signatures resulted from interaction model 1, which uses CD4^+^ gene expression change (Day 28/Day 0 gene expression change), rather than Day 0 gene expression or Day 28 gene expression.

**Table 4 viruses-14-02763-t004:** Enriched biological pathways/processes involved in the CD4+ T cell activation by vaccine type (from interaction model 1: Day 28/Day 0 gene expression effect on immune outcome by vaccine type).

KEGG Pathway *	Set Size	*p*-Value	FDR ***
Herpes simplex virus 1 infection	388	1.48 × 10^−6^	0.001
MAPK signaling pathway **	150	3.11 × 10^−6^	0.001
Endocytosis	142	2.60 × 10^−5^	0.003
Human papillomavirus infection	158	3.77 × 10^−5^	0.003
Thyroid hormone signaling pathway	69	4.57 × 10^−5^	0.003
Kaposi sarcoma-associated herpesvirus infection	101	3.19 × 10^−4^	0.011
Tight junction **	81	3.38 × 10^−4^	0.011
C-type lectin receptor signaling pathway	56	3.54 × 10^−4^	0.011
Folate biosynthesis	14	4.15 × 10^−4^	0.011
PD-L1 expression and PD-1 checkpoint pathway in cancer	45	4.18 × 10^−4^	0.011
Glucagon signaling pathway	51	4.18 × 10^−4^	0.011
Rap1 signaling pathway **	100	4.56 × 10^−4^	0.011
Relaxin signaling pathway	55	4.59 × 10^−4^	0.011
TNF signaling pathway **	62	6.22 × 10^−4^	0.013
GnRH signaling pathway	44	6.59 × 10^−4^	0.013
NOD-like receptor signaling pathway **	95	9.54 × 10^−4^	0.017
Hepatitis B	84	1.08 × 10^−3^	0.018
Glycosaminoglycan biosynthesis—chondroitin/dermatan sulfate	16	1.24 × 10^−3^	0.019
Notch signaling pathway **	40	1.34 × 10^−3^	0.020
Human T-cell leukemia virus 1 infection	128	1.45 × 10^−3^	0.020
Glutamatergic synapse	44	1.67 × 10^−3^	0.022
IL-17 signaling pathway **	40	1.78 × 10^−3^	0.023
Transcriptional misregulation in cancer	103	2.01 × 10^−3^	0.025
Human cytomegalovirus infection	112	2.52 × 10^−3^	0.028
Epstein-Barr virus infection	114	2.66 × 10^−3^	0.028
Phospholipase D signaling pathway	69	2.72 × 10^−3^	0.028
Viral carcinogenesis	110	2.96 × 10^−3^	0.029
Calcium signaling pathway **	90	2.96 × 10^−3^	0.029
Neutrophil extracellular trap formation	87	3.05 × 10^−3^	0.029
Estrogen signaling pathway	56	3.14 × 10^−3^	0.029
AGE-RAGE signaling pathway in diabetic complications	52	3.50 × 10^−3^	0.030
Growth hormone synthesis, secretion, and action	55	4.56 × 10^−3^	0.036
Circadian entrainment	39	4.63 × 10^−3^	0.036
Serotonergic synapse	41	4.94 × 10^−3^	0.038
Insulin signaling pathway	72	5.22 × 10^−3^	0.039
RNA polymerase	29	6.12 × 10^−3^	0.044
Inflammatory mediator regulation of TRP channels **	49	6.28 × 10^−3^	0.044
Fatty acid elongation	16	6.55 × 10^−3^	0.045
Th1 and Th2 cell differentiation **	50	7.04 × 10^−3^	0.046
Necroptosis	76	7.04 × 10^−3^	0.046
Dopaminergic synapse	55	7.21 × 10^−3^	0.047
Apoptosis **	90	7.35 × 10^−3^	0.047

* From the enriched pathways list, we removed pathways associated with diseases that arose from combinations of other pathways. ** Pathways related to T cell differentiation, activation, and function are bolded. *** FDRs were calculated using Benjamini–Hochberg method (Methods).

## Data Availability

All data included in the current study are available from the corresponding author upon reasonable request.

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
