# Peer review of "T Cell Transcriptional Signatures of Influenza A/H3N2 Antibody Response to High Dose Influenza and Adjuvanted Influenza Vaccine in Older Adults"

_viruses, 2022, doi:10.3390/v14122763_

Round 1
Reviewer 1 Report
The manuscript entitled ‘T cell Transcriptional Signatures of Antibody Response to High Dose Influenza and Adjuvanted Influenza Vaccine in Older Adults’ by Haralambieva et. al. focuses on identifying the transcriptional signature of CD4+ T cells in older adults inoculated with high dose (Fluzone) and adjuvanted (Fluad) influenza vaccines and aims to find the relationship of these signatures with humoral immune responses. The manuscript is nicely written but I have few suggestions to improve it.
1- It will be ideal that authors include a scatter plot or volcano plot to highlight the perturbed genes in the 3 interaction models.
2- Authors have performed RNA-seq for CD4+ T cells and they identified 128 genes that are perturbed after vaccination (Day 28/Day 0). However, Table 2 does not have all these genes. Same is the case for Table 3 and 4.
3- There is no explanation given that why 4 genes (CAMK4, TMEM38B, CREBBP, GATA3) were chosen for further analysis. They don’t seem to be top genes. What is the result of analysis performed on all differentially regulated 128 genes in interaction model 1?
4- All the genes analyzed have negative association with the immune outcome in case of Fluad vaccine, while positive association between these two parameters was observed in case of Fluzone vaccine. How do authors explain these results?
5- In Figure 2, Fluad samples are represented with open circle, but legend shows closed circle.
Author Response
Dear Reviewer,
Thank you very much for your time reviewing our manuscript and your constructive comments and suggestions. We strongly believe that your comments and suggestions greatly strengthen this manuscript. We have considered your comments/suggestions in depth and responded to each of your comments/suggestions in detail as below. We have also revised the manuscript accordingly and uploaded the revised manuscript for your reference. All changes in the revised manuscript were tracked. We have also uploaded our responses to the comments/suggestions from other reviewers for your information.
Please do not hesitate to contact me if any questions arise.
Sincerely,
Richard B. Kennedy, Ph.D.

Reviewer 2 Report
In this manuscript, the authors identified genes/proteins and pathways essential for cell activation and function in CD4+ T cells that are associated with differences in influenza vaccine-induced humoral immunity by vaccine type. The authors try to explore the mechanism of declining influenza vaccine-induced immunity in older people. The immunity of vaccine in older people is significant in the research area, while which is very complex to uncover the mechanism.
There are some major issues need to be resolved
1. HAI titer of H3N2 could not represent the whole humoral immunity of trivalent vaccine. Meanwhile the humoral immunity of H1N1 and B/Colorado/06/2017-like (Victoria lineage) in the trivalent vaccines Fluzone and Fluad should be tested and analyzed. Transcriptional signatures of H1N1 and B-victoria strain antibody should be assessed,analyzed and compared to data of H3N2.
2. The authors mentioned, ‘Collectively, our study identified important T cell factors and functions regulating influenza vaccine humoral immunity by vaccine type. Further validation and new analyses regarding these unique data are in progress, including modeling of cellular immune response outcomes following vaccination.’
In this manuscript, transcriptional signatures of T cell response should be included. CAMK4 gene, solute carrier family proteins and other pathway genes are identified in the models, most of those genes are associated with T cell response. Cellular immune response to fluad and Fluzone vaccination is essential to understand the immunity in older people after vaccination. Th1 related cytokines (IFN-γ, IL-2 or TNF-α) could be detected by Elispot or FACS, further the transcriptional signatures of T cell response analysis would be helpful for understanding immunity mechanism in vaccinated older people.
3. The authors should explain the totally different signatures of humoral immune response to fluad and Fluzone vaccination. Is the effect related to antigen dose of vaccines or adjuvant addition?
4. In model 1, the author identified Th2 T cell differentiation regulator GATA binding protein 3, which are associated to Th1/Th2 cell differentiation. It will be interesting to investigate transcriptional signatures of Th1/Th2 response by Th1 and Th2 typing experiment in the PBMCs.
5. Nakaya, H.I et al reported systems biology of immunity to MF59-adjuvanted versus nonadjuvanted trivalent seasonal influenza vaccines in in 14- to 24-months old children. It is necessary to discuss gene/protein signature differences in humoral immunity following adjuvant and non-adjuvant vaccines in children and older adults. Both the manuscript and Nakaya’s reports investigated the transcriptional or genetic signature of humoral immunity, it would be interesting to analyze and discuss function of adjuvants in elder and children.
6. This manuscript showed CAMK4 is associated with influenza vaccine-induced humoral immunity, actually CAMK4 was broadly reported that it is negative regulator of influenza vaccine humoral immunity. In this manuscript, function of CAMK4 is consistent with previous reports after fluad vaccination, but CAMK4 is a positive regulator in Fluzone vaccination. The mechanism for the above differences should be discussed.
More problematic issues
It is reasonable RNA is derived from CD4+ T cells in figure 1. But in Lines 116-117, the author descripted total RNA was extracted from 1×106 cryopreserved PBMCs,
CD4+ T cells information such as separation from PBMCs, stock and CD4+ mRNA extracted should be mentioned in the methods.
Author Response

(The authors gave the same response as above.)

Round 2
Reviewer 1 Report
Authors have addressed the concerns.
Author Response
December 06, 2022
RE: Manuscript #Viruses-2015129 entitled “T cell Transcriptional Signatures of Influenza A/H3N2 Antibody Response to High Dose Influenza and Adjuvanted Influenza Vaccine in Older Adults”.
Dear Reviewer,
Thank you very much for your time reviewing our manuscript. Since you did not have any specific comments/request for the revision, we uploaded the revised manuscript for your reference. As suggested by the reviewer 2 and academic editor, we have revised the title of the manuscript. Please kindly check the revised manuscript and do not hesitate to contact me if any questions arise.
Sincerely,
Richard B. Kennedy, Ph.D.

Reviewer 2 Report
The revised version was greatly improved, i am satisfied to all the responses except for #1.
I agree H3N2 caused higher mortality in older adults, but WHO recommend four influenza strains every year, the other three strains also effect immunity of older people. Since the authors have been starting to test HAI responses to the other vaccine strains, I think it only need take several days to complete all these HAI assays, due to the PBMC samples of patients are there. In the manuscript, all the trivalent strains' humoral immunity results are required, otherwise the title should be revised.
Author Response
December 06, 2022
RE: Manuscript #Viruses-2015129 entitled “T cell Transcriptional Signatures of Influenza A/H3N2 Antibody Response to High Dose Influenza and Adjuvanted Influenza Vaccine in Older Adults”.
Dear Reviewer,
Thank you very much for your time reviewing our manuscript and your constructive comments and suggestions. We strongly believe that your comments and suggestions greatly strengthen this manuscript. We have considered your comments/suggestions in depth and revised the title of the manuscript as your suggestion. All changes in the revised manuscript were tracked.
Please do not hesitate to contact me if any questions arise.
Sincerely,
Richard B. Kennedy, Ph.D.
